# Synthesis, Characterization, Photocatalysis, and Antibacterial Study of WO_3_, MXene and WO_3_/MXene Nanocomposite

**DOI:** 10.3390/nano12040713

**Published:** 2022-02-21

**Authors:** Al-Zoha Warsi, Fatima Aziz, Sonia Zulfiqar, Sajjad Haider, Imran Shakir, Philips O. Agboola

**Affiliations:** 1Baghdad-ul-Jadeed Campus, Institute of Chemistry, The Islamia University of Bahawalpur, Bahawalpur 63100, Pakistan; zohawarsi2208@gmail.com (A.-Z.W.); fatimaaziz408@gmail.com (F.A.); 2Department of Chemistry, School of Sciences & Engineering, The American University in Cairo, New Cairo 11835, Egypt; sonia.zulfiqar@aucegypt.edu; 3Department of Chemical Engineering, College of Engineering, King Saud University, P.O. Box 800, Riyadh 11421, Saudi Arabia; shaider@ksu.edu.sa; 4Sustainable Energy Technologies (SET) Center, College of Engineering, King Saud University, P.O. Box 800, Riyadh 11421, Saudi Arabia; mshakir@ksu.edu.sa; 5College of Engineering, Al-Muzahmia Branch, King Saud University, P.O. Box 800, Riyadh 11421, Saudi Arabia

**Keywords:** WO_3_, MXene, XRD, FESEM, EDX, photocatalysis

## Abstract

Tungsten oxide (WO_3_), MXene, and an WO_3_/MXene nanocomposite were synthesized to study their photocatalytic and biological applications. Tungsten oxide was synthesized by an easy and cost-effective hydrothermal method, and its composite with MXene was prepared through the sonication method. The synthesized tungsten oxide, MXene, and its composite were characterized by X-ray diffraction (XRD), field emission scanning electron microscopy (FESEM), Fourier transform infrared (FTIR), energy-dispersive X-ray analysis (EDX), and Brunauer–Emmett–Teller (BET) for their structural, morphological, spectral, elemental and surface area analysis, respectively. The crystallite size of WO_3_ calculated from XRD was ~10 nm, the particle size of WO_3_ was 130 nm, and the average thickness of MXene layers was 175 nm, which was calculated from FESEM. The photocatalytic activity of as-synthesized samples was carried out for the degradation of methylene blue under solar radiation, MXene, the WO_3_/MXene composite, and WO_3_ exhibited 54%, 89%, and 99% photocatalytic degradation, respectively. WO_3_ showed maximal degradation ability; by adding WO_3_ to MXene, the degradation ability of MXene was enhanced. Studies on antibacterial activity demonstrated that these samples are good antibacterial agents against positive strains, and their antibacterial activity against negative strains depends upon their concentration. Against positive strains, the WO_3_/MXene composite’s inhibition zone was at 7 mm, while it became 9 mm upon increasing the concentration. This study proves that WO_3_, MXene, and the WO_3_/MXene nanocomposite could be used in biological and environmental applications.

## 1. Introduction

In the last few decades, environmental remediation technologies have been the most challenging for effective and efficient water cleaning, primarily through the photocatalytic method [1,2,3,4,5,6]. Such catalysts are cost-effective and have a suitable energy and electronic structure. Minimal amounts of contaminants such as phenols, textile dyes, and poly chlorinated biphenyls (PCBs) not only pollute the water, but also reduce the growth of aqueous organisms [7]. For the removal of such pollutants from water, various physical and chemical methods were reported [8,9]. However, these methods are either expensive or suitable for large amounts of contaminants [10]. So, the photocatalytic degradation of organic contaminants has gained much attention due to its efficiency and cost-effectiveness [7,11].

Owing to their surface, electronic and crystal structure, various semiconductors, mainly metal oxides such as ZnO, TiO_2_, Fe_2_O_3_ and WO_3_, and sulfides (ZnS, CdS) showed exceptional photocatalytic behavior [10,12,13]. Among all of these metal oxides, TiO_2_ exhibited good photostability and photocatalytic activity in an aqueous medium [14,15]. However, due to its high charge recombination and wide band gap, its applications are restricted [16]. Meanwhile, Fe_2_O_3_ and ZnO are quite unstable in wastewater at various pH values, which limits its applications [10]. Sulfide-based catalysts are also not suitable for water remediation because they release toxic sulfides on illumination [10]. Among several metal oxides, WO_3_ is a more appropriate candidate for photocatalytic degradation owing to its abundance and narrow band gap, physical and chemical stability, and photosensitivity in visible-light area [17]. Among major global problems, cancer is the most important, and scientists are striving to solve it [18,19]. Recently, they succeeded in tracking cancer using tungsten oxide in mice [20]. So, the preparation of tungsten oxide is quite interesting. So far, various technologies have been used and reported for the synthesis of tungsten oxide, such as hydrothermal method, acid precipitation, and sol-gel [21,22,23,24,25,26,27]. By controlling the reaction time, precursor material, hydrothermal temperature, and capping agents, the optical properties, crystallinity, and morphology of WO_3_ nanostructures can be varied. MXene, an emerging 2D material, is a layered structured transition metal, nitride or carbide, having both a hydrophilic nature and high electrical conductivity. They have large interlayer spacing, a greater surface area, and a large number of active surface sites [28]. They can also sequester and remove dyes, heavy metals, and radioactive nuclides [29].

MXene is a potential candidate for the synthesis of electrode material for various energy storage devices such as supercapacitors and batteries. Metal-oxide- and metal-sulfide-based composites were reported to be better electrode materials for supercapacitor electrodes [30,31,32,33,34,35,36,37,38]. Therefore, MXene-based composites were also explored for better properties [39,40]. The capacitance of MXene-based devices can be further enhanced by producing its composites with other materials such as reduced graphene oxide, metal oxides, and conducting polymers [41,42]. Scientists are also attempting to use certain pristine metal oxides and their composites with 2D materials in various biological applications [43,44,45,46,47].

The main goal behind this research is to develop a unique WO_3_/MXene composite that exhibits potential applications in biological and environmental remediation. In this paper, MXene was synthesized by etching an Al atomic layer from MAX powder. The hydrothermal route was used to synthesize tungsten oxide nanorods. The WO_3_/MXene composite was synthesized by a simple sonication method. These prepared samples were characterized for structural, spectral, morphological, and elemental analyses. The photocatalytic and antibacterial activity of the as-synthesized samples was measured and is discussed in detail. 

## 2. Experimental Work

### 2.1. Materials

MAX powder (Ti_3_AlC_2_) (98% purity); hydrofluoric acid (HF) (40 wt %, Merck, Darmstadt, Germany); deionized (DI) water; sodium tungstate (Na_2_WO_4_.2H_2_O, 99%, Sigma-Aldrich, Burlington, MA, USA), sodium sulfate (Na_2_SO_4_, 99%, Sigma-Aldrich, Burlington, MA, USA), HCl (36%, Fischer Scientific, Waltham, MA, USA).

### 2.2. Synthesis of MXene

MAX powder (Ti_3_AlC_2_) was used to prepare the MXene with the Ti_3_C_2_T_x_ formula in a 50 mL Teflon vessel. For this purpose, Al was etched by using an HF solution. For the preparation of MXene, 10 mL HF was poured inside the Teflon vessel and then placed in a fume hood. Then, 0.5 g of MAX powder was slowly added into the HF solution pinch by pinch. Then, the whole mixture was stirred magnetically at room temperature for about 24 h for maximal etching. DI water was added to the resultant product for dilution, and multilayered MXene was obtained by centrifugation at 5000 rpm. The washing of these precipitates was repeated continuously until its pH became 6. The vacuum filtration of the aqueous dispersion was carried out by using a PTFE membrane. The filtrate containing Ti_3_C_2_T_x_ was then freeze-dried for 24 h. Schematic illustration for preparation of MXene is shown in Figure 1. 

### 2.3. Synthesis of Tungsten Oxide (WO_3_)

2.5 g of sodium tungstate and 3.0 g of sodium sulfate were dissolved in 80 mL of distilled water. A 3M HCl solution was added dropwise to the clear solution under continuous stirring, and the pH of the solution was set to 1.5. After 10 min of stirring, the mixture was transferred into a Teflon-lined stainless-steel autoclave and was kept at 180 °C for 48 h. After that, the product was collected by centrifugation at 4500 rpm, and washed with distilled water and ethanol to obtain neutral solution; then, the product was obtained by drying at 60 °C in air.

### 2.4. Synthesis of WO_3_/MXene Composites

The composite of WO_3_/MXene (1:1) was fabricated by sonication method. Then, 2 g of MXene was added in 50 mL of water and sonicated for 3 h. Afterwards, 2 g of tungsten oxide was added to it, again sonicated for 2 h, and then dried in an oven. Synthesis of WO_3_ and WO_3_/MXene composites is shown in Figure 2.

### 2.5. Characterization

An XRD diffractometer using Cu Kα radiation (λ = 1.54 Å) as a light source, at a scan rate of 30 min by applying a voltage of 40 kV, was used for the structural and phase analysis of the as-synthesized samples. ZEISS LEO SUPRA 55 field emission scanning electron microscope and JEOL JCM-6000Plus SEM were used for morphological characterization and elemental analysis, respectively. Functional group analysis and the surface properties of the as-synthesized samples were measured by Fourier transform infrared spectroscopy (FTIR). For the measurement of the BET surface areas, nitrogen adsorption–desorption was conducted by flowing liquid nitrogen at 77 K (−196 °C) by using the Micromeritics ASAP 2020 Physisorption analyzer.

### 2.6. Photocatalytic Degradation

WO_3_, MXene, and the WO_3_/MXene nanocomposite were used as a photocatalyst to measure the photocatalytic degradation of methylene blue in the presence of solar radiation for 80 min. For these measurements, 100 mL of 5 ppm methylene blue solution was poured in a beaker, and 5 mg of photocatalyst was added into the solution. It was then stirred continuously for 60 min in the dark. Adsorption–desorption equilibrium could thus be achieved between methylene blue and photocatalyst. The solution was then placed in solar light with constant stirring. In order to measure the degradation percentage of methylene blue, 5 mL of a solution containing both dye and sample was taken after every 10 min, and a UV–vis spectrophotometer was used to measure the degradation efficiency of the samples [5,48].

The degradation percentage of the as-synthesized samples was measured by using following equation:(1)$$\;\%degradation=\frac{\left(C_{0}-C_{t}\right)}{C_{0}}\times 100\;$$
where, *C_t_* is the concentration of the solution at time *t*, and *C*_0_ is the concentration of the solution at time zero.

### 2.7. Antibacterial Activity

The disc diffusion method was utilized to study the antibacterial activity of WO_3_, MXene, and WO_3_/MXene nanocomposite. *Staphylococcus aureus* (*S. aureus*) was used as a positive strain, and *Escherichia coli* (*E. coli*), *Klebsiella pneumonia* (*K. pneumonia*) and *Proteus vulgaris* (*P. vulgaris*) were used as negative strains. For standard/positive control, an antimicrobial agent (ciprofloxacin) was used. First, the aqueous solution of the as-prepared samples was prepared by sonicating the samples with distilled water. Then, they were placed on the corners of a nutrient agar plate with the use of forceps. After incubating the samples for 24 h at 37 °C, the zone of inhibition could be seen on the edges of the agar plate. The formation of these zones of inhibition confirmed the antibacterial activity, while the lack of these zones of inhibition showed no antibacterial activity. The mm units were used for the measurement of these inhibition zones.

## 3. Results and Discussion

### 3.1. XRD

The different phases of the as-fabricated WO_3_, MXene, and their composite (WO_3_/MXene) were studied by utilizing the Cu Kα radiation (λ = 1.5406 Å) with an X-ray diffractometer. Figure 3 shows the X-ray diffractograms. Tungsten oxide (WO_3_) produced the diffraction peaks at 2 theta values 23°, 26°, 33°, 41°, 49° and 55°. The Miller indices corresponding to these peaks are (001), (101), (111), (110), (220) and (202) [49,50]. At 2θ = 23°, tungsten oxide (WO_2.95_) gave a characteristic peak. For WO_3_, peaks were more prominent at 2θ = 26°, corresponding to the Miller indices (111) [51]. The structure of the as-prepared WO_3_ nanoparticles was compared with JCPDS card 00-002-0310. 

The pure MAX XRD pattern showed 2θ peaks at 9.11°, 18.7°, 33.58°, 35.65°, 38.61°,41.44°, 51.93°, 56.08°, and 60.23°, which corresponds to Miller indices (002), (004), (101), (103), (104), (105), (108), (109), and (110), respectively [52,53]. Due to the presence of the Al, the pure MAX powder showed a characteristic peak at 2θ = 38.61°, which corresponds to (104). Al was completely etched by using HF in order to fabricate good-quality MXene [54,55]. During the first 2 h of the reaction, the peak intensity at 38.61° increased [56]. After 24 h of the reaction, the characteristic peak of MAX at (104) vanished, as shown in Figure 3a. A peak shift was also observed in the peak at 9.11° [57]. 

The XRD pattern was used for studying the phases of the composite (WO_3_/MXene). Figure 3c shows the XRD pattern of the composite (WO_3_/MXene), which possessed diffraction peaks at 23°, 26°, 33°, 41°, 49°, 55° and 60.23° corresponding to Miller indices (001), (101), (111), (110), (220), (202) and (110), respectively. All these peaks included almost all the specific peaks of tungsten and MXene, and no additional peak was observed in the case of composite.

By using the Debye–Scherer equation, the crystallite size of the as-fabricated tungsten oxide was calculated [58].
*D* = *Kλ*/*βCosθ*(2)
where *D* is the crystalline size; *K* is the Scherer constant; λ is the X-ray wavelength of the copper source used in XRD, which was equal to 1.5406 Å; Bragg’s angle was given by *θ*; and *β* represents full width at half maximum (FWHM) [59]. The crystalline size of WO_3_ nanoparticles, determined by XRD, was 6.19 nm. The measurement of the crystalline size of MXene was not possible by using the Debye–Scherer formula because MXene is a 2D layered material.

### 3.2. FESEM and EDX Analysis

For FESEM analysis, the samples were gold-sputtered for 120 s at 15 mA before imaging. Figure 4a,b show the morphology of WO_3_ and WO_3_/MXene nanocomposite, respectively. Figure 4a demonstrates the block-/rodlike morphology of WO_3_. Figure 4b clearly shows that MXene was impregnated on the nanorods of WO_3_. The nanosheet-like structure in Figure 4c represents the formation of MXene. The particle size of WO_3_ was ~130 nm, which was calculated from the FESEM image. The average layer thickness of MXene calculated from micrograph was ~175 nm.

Energy-dispersive X-ray analysis (EDX) was used for the elemental analysis of the synthesized material. Figure 5a,b show the elemental composition of WO_3_ and WO_3_/MXene composites, respectively, which confirmed the purity of the as-synthesized samples.

### 3.3. FTIR

FTIR spectroscopy was used for the spectral analysis of the samples, which indicates the composition of synthesized products. Figure 6 shows the FTIR spectra of MXene, WO_3_ and WO_3_/MXene nanocomposite. In the case of MXene, the absorption band present at around 3545 cm^−1^ was attributed to the absorbed water, which was due to the hydrophilic nature of MXene [60]. The bands present in the range of 2000–2500 cm^–1^ showed a methyl/methylene group (–CH_3_, CH_2_). The signals at 603 and 1529 cm^−1^ were characteristic of Ti–O and C–F, respectively. The FTIR spectrum of WO_3_ featured characteristics bands of W–O–W and W–O at around 735 and 836 cm^−1^ [49]. The spectrum of the WO_3_/MXene nanocomposite showed the absorption bands of both MXene and WO_3_.

### 3.4. BET Measurements

Average particle size, BET surface area, total pore volume, and average pore width were determined from nitrogen adsorption-desorption curves (Figure 7) and their values are given in Table 1. From the BET results, it was predicted that the formation of the composite of WO_3_ with MXene would result in increased surface area and enhanced average pore width, while average particle size was reduced. The reason behind this is the 2D layer structure of MXene, which offers a greater surface area. However, the photocatalytic activity of WO_3_ was higher than that of the composite because MXene only enhanced the surface area, but this increased surface area had no effect on the degradation of dyes because the adsorption capacity and band gap of MXene were much less, due to which charge separation was not effective.

### 3.5. Photocatalysis

The photocatalytic activity of WO_3_, MXene, and the WO_3_/MXene nanocomposite was measured for the degradation of methylene blue under solar radiation for 80 min. The initial concentration of methylene blue was determined by measuring the blank absorption of the dye solution. For the achievement of adsorption–desorption equilibrium between photocatalyst and methylene blue, the solution was placed in the dark for 1 h with continuous stirring. The solution containing both methylene blue and sample was then kept under solar radiation. By taking 5 mL solution after regular intervals, the degradation of the dye was measured by using a UV–vis spectrophotometer [61].

The absorption spectra of methylene blue using WO_3_, MXene, and the WO_3_/MXene nanocomposite as photocatalyst are shown in Figure 8a–c). For the description of the experimental data given in Figure 9, a pseudo-first-order model was utilized, and the values of K measured by this model were 0.05682, −0.0084, and 0.0346 for WO_3_, MXene, and the WO_3_/MXene nanocomposite, respectively.

Figure 10 demonstrates the removal efficiency of WO_3_, MXene, and the WO_3_/MXene nanocomposite. WO_3_ showed higher degradation ability as compared to that of MXene and the WO_3_/MXene composite. The reason behind this high photocatalytic activity is the greater band gap of WO_3_, which allowed for them to absorb a wide-spectrum range of sunlight and degrade the dye solution with this solar energy. MXene exhibited very low removal efficiency, while the degradation ability of WO_3_/MXene composite was between those of WO_3_ and MXene. MXene is a 2D material that acts as a supporting material. WO_3_ is material that involves the generation of photo produced electrons and holes. MXene merely increases the surface area and reduces the chances of recombination of these photogenerated electrons and holes. Figure 11 shows the comparison of the degradation percentage of methylene blue by WO_3_, MXene, and the WO_3_/MXene nanocomposite.

#### Mechanism

An emerging degradation technology that leads to the removal of most contaminants is heterogeneous photocatalysis [44]. The comparison of current reported catalysts with already reported similar materials is given in Table 2. 

The proposed mechanism involved in photocatalytic degradation consists of the following steps [74] and also depicted in Figure 12:Efficient photons from sunlight are absorbed by *WO*_3_:
(3)$$\left(WO_{3}\right)+h\upsilon \;\to \;e_{CB}^{-}+h_{VB}^{+}\;$$

2.Ion sorption of oxygen takes place (start of oxygen reduction where the oxidation state of oxygen changes from 0 to −1/2).


(4)
$$\;{\left(O_{2}\right)}_{ads}+e_{CB}^{-}\;\to \;O_{2}^{.-}\;$$


3.Photogenerated holes neutralize the *–OH* group and produce *OH*° radicals.


(5)
$${\left(H_{2}O\;↔\;h^{+}+OH^{-}\right)}_{ads\;}+h_{VB}^{+}\;\to \;H^{+}+OH^{{}^{\circ}}$$


4.Protons neutralize the ${O_{2}^{{}^{\circ}}}^{-}$


(6)
$${O_{2}^{{}^{\circ}}}^{-}+H^{+}\;\to \;HO_{2}^{{}^{\circ}}$$


5.Dismutation of oxygen occurs, and transient *H*_2_*O*_2_ is formed:


(7)
$$\;2H{O_{2}^{{}^{\circ}}}^{-}\;\to \;H_{2}O_{2}+O_{2\;}$$


6.Oxygen is reduced for the second time, and the decomposition of *H*_2_*O*_2_ occurs:


(8)
$$H_{2}O_{2}+e^{-}\;\to \;OH^{{}^{\circ}}+OH^{-}$$


7.*OH*° radical attacks the organic pollutant (*dye*) and ultimately causes its oxidation:


(9)
$$\;Dye+OH^{{}^{\circ}}\;\to \;Dye^{{}^{\circ}}+H_{2}O$$


8.Direct oxidation takes place when it reacts with holes:


(10)
$$R+h^{+}\;\to \;{R^{+}}^{{}^{\circ}}\;\to Degradation\;products\;$$


### 3.6. Antibacterial Activity

Among metal oxide nanoparticles, ZnO is a competitive candidate for the study of antibacterial activity. Recent studies showed that ZnO nanoparticles could activate endoplasmic reticulum stress and ultimately kill mammalian cells [75]. Therefore, scientists have been striving to explore new nano-antibacterial agents with better compatibility. Recently, WO_3__−x_ was verified to exhibit good biocompatibility and antibacterial activity [76]. In the current study, WO_3_, MXene, and the WO_3_/MXene nanocomposite were used as antibacterial agents for the study of antibacterial activity (Figure 13, Figure 14, Figure 15 and Figure 16). Table 3 shows the zones of inhibition of WO_3_, MXene, and the WO_3_/MXene nanocomposite. The disc diffusion method was utilized to measure the inhibition zones of the as-prepared samples, and various positive strains (*S. aureus*) and negative strains *E. coli*, *K. pneumonia* and *P. vulgaris* were used for antibacterial activity measurements. Due to the structural differences of cell membranes and cell walls, the as-synthesized samples exhibited different sensitivity levels towards the positive and negative strains [77,78]. Table 3 shows that, with the positive strain (*S. aureus*), all samples showed good antibacterial activity, which increased with the increase in concentration. In the case of negative strains, all samples were active against *K. pneumoniae*, and the WO_3_/MXene composite showed good activity at a low concentration. When the concentration of MXene and WO_3_ increased, activity also increased. The WO_3_/MXene nanocomposite showed no activity against *E. coli* and *P. vulgaris*, while WO_3_ and MXene exhibited good antibacterial activity, which was enhanced on the increase in concentration. The reason behind the low or zero antibacterial activity of the WO_3_/MXene composite against negative strains was the presence of an extra outer membrane that increased the resistance of Gram-negative strains to WO_3_/MXene. The WO_3_/MXene nanocomposite showed a decrease in antibacterial activity on an increase in concentration due to certain factors such as size and agglomeration. Due to these factors, these nanocomposites were not able to penetrate the bacterial cell wall; hence, its toxicity decreased. On the other hand, the pristine WO_3_ and MXene showed an increase in antibacterial activity on increasing concentration.

### 3.7. Mechanism of Antibacterial Activity

The mechanism involved with the antibacterial activity of the as-synthesized nanoparticles was the cell damage by electrostatic interactions between the cell membrane and metal oxide nanoparticles. The main sites of attraction of metal cations are the chemical groups of polymers on membranes of bacteria that are electronegative in nature. The carboxylic groups present in the proteins are the main reason behind the negative charge on the surface of bacteria. Electrostatic attraction is created due to the charge difference between bacterial membrane and metal oxide nanoparticles; thus, these nanoparticles accumulated on the cell surface and ultimately entered the bacteria. This interaction between membrane polymer and cationic metal oxide nanoparticles resulted in the cytoxicity of microorganisms. The available surface area and ratio of particle size to surface area determine the efficiency of metal oxide nanoparticles in bacterial growth inhibition. The permeability and structure of the cell membrane are changed due to the attachment of metal oxide nanoparticles. The disorganization of cell wall was due to the strong bond between positively charged metal oxide nanoparticles and membrane. Apart from binding with the cell membrane, these metal oxide nanoparticles also bind with mesosomes, resulting in the alteration of cell division, DNA replication, and cellular respiration [79].

## 4. Conclusions

In the current work, we prepared WO_3_, MXene, and a WO_3_/MXene nanocomposite, which exhibited their potential applications in the biological and environmental remediation fields. WO_3_, MXene, and the WO_3_/MXene nanocomposite were synthesized by hydrothermal method, wet chemical etching, and sonication method, respectively. XRD, FTIR, EDX, and FESEM were used to characterize the as-synthesized samples for structural, spectral, elemental, and morphological analysis, respectively. BET analysis was conducted for surface area determination. The photocatalytic degradation of methylene blue using WO_3_, MXene, and the WO_3_/MXene nanocomposite was 99%, 54%, and 89%, respectively. The photocatalytic activity of WO_3_ was significant. MXene is a 2D material, its photocatalytic activity is very low, and it only acted as supporting material by enhancing the photocatalytic ability of its composite with WO_3_. The as-prepared samples also exhibited good antibacterial activity against positive strain bacteria; in the case of negative strains, WO_3_, MXene, and the WO_3_/MXene nanocomposite exhibited antibacterial activity at high concentrations.

## Figures and Tables

**Figure 1 nanomaterials-12-00713-f001:**
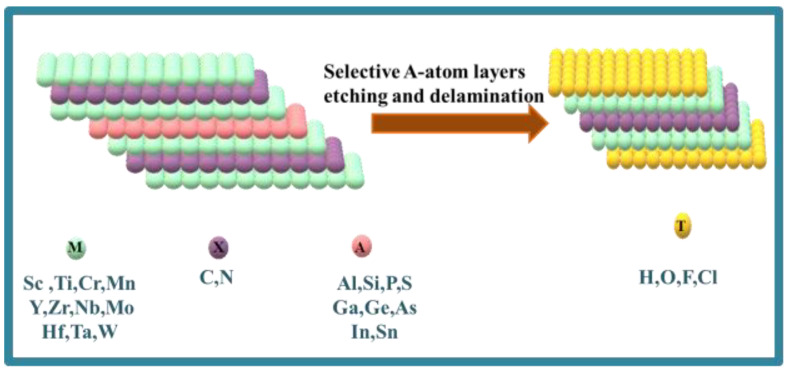
Structural representation of MXene.

**Figure 2 nanomaterials-12-00713-f002:**
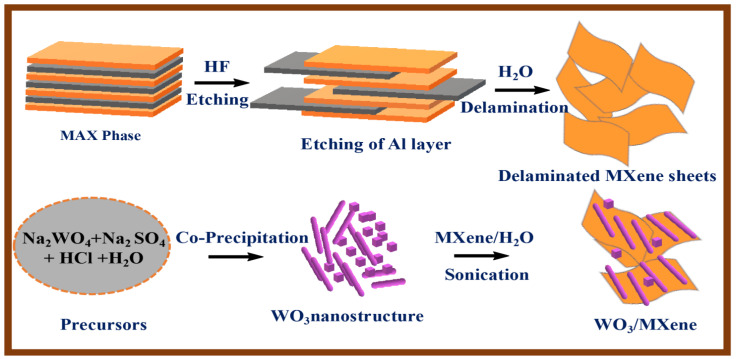
Schematic illustration summarizing synthesis of MXene, WO_3_, and WO_3_/MXene nanocomposite.

**Figure 3 nanomaterials-12-00713-f003:**
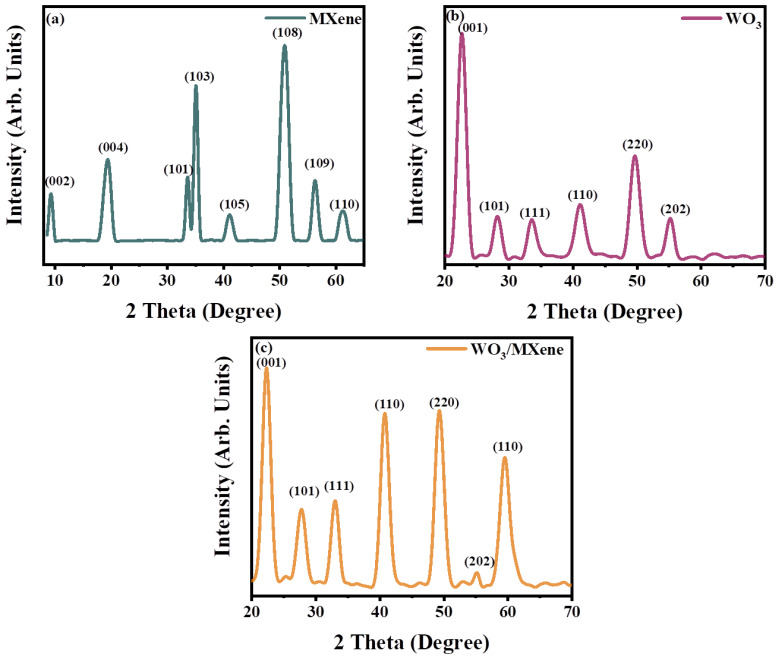
XRD spectra of (**a**) MXene, (**b**) WO_3,_ and (**c**) WO_3_/MXene nanocomposite.

**Figure 4 nanomaterials-12-00713-f004:**
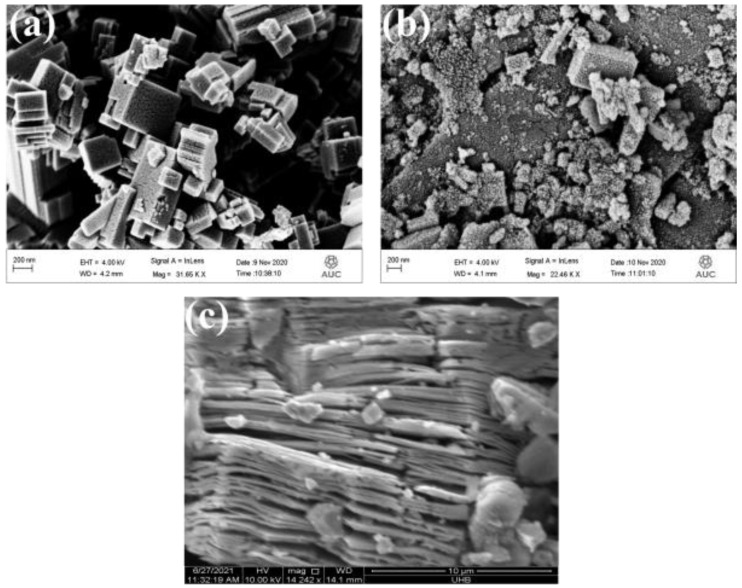
FESEM images of (**a**) WO_3_, (**b**) WO_3_/MXene nanocomposite and (**c**) MXene.

**Figure 5 nanomaterials-12-00713-f005:**
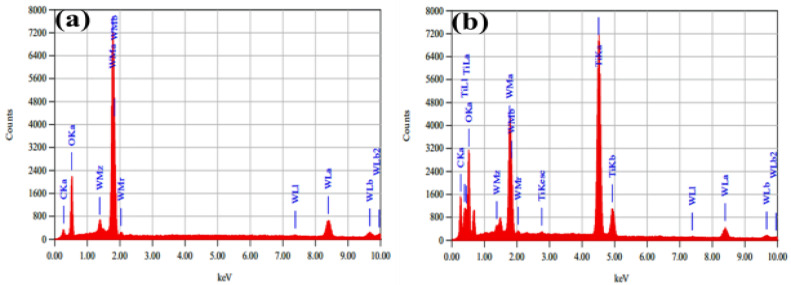
EDX analysis of (**a**) WO_3_ and (**b**) WO_3_/MXene nanocomposite.

**Figure 6 nanomaterials-12-00713-f006:**
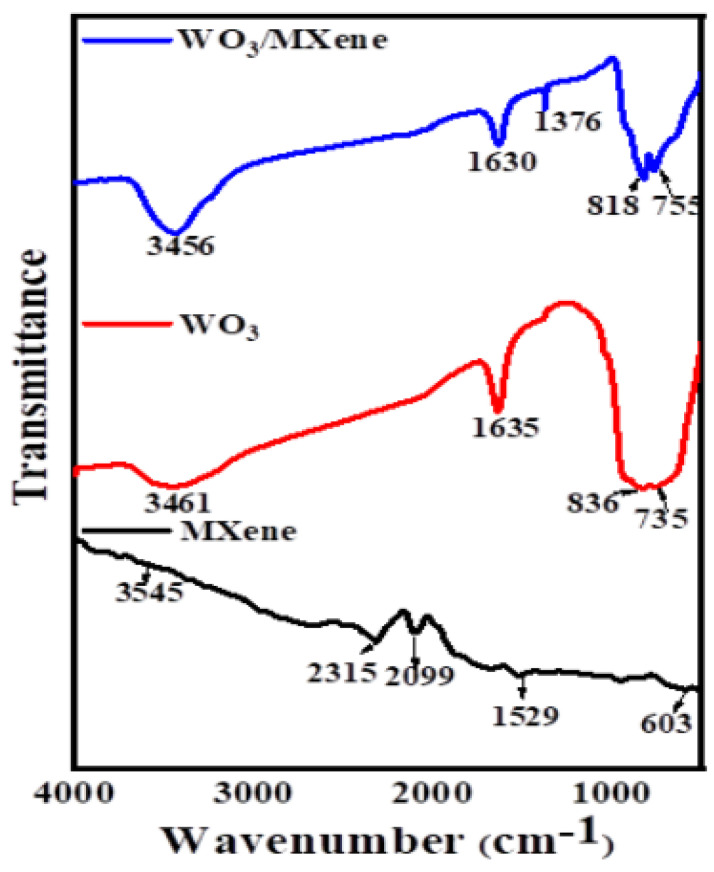
FTIR spectra of MXene, WO_3_, and WO_3_/MXene nanocomposite.

**Figure 7 nanomaterials-12-00713-f007:**
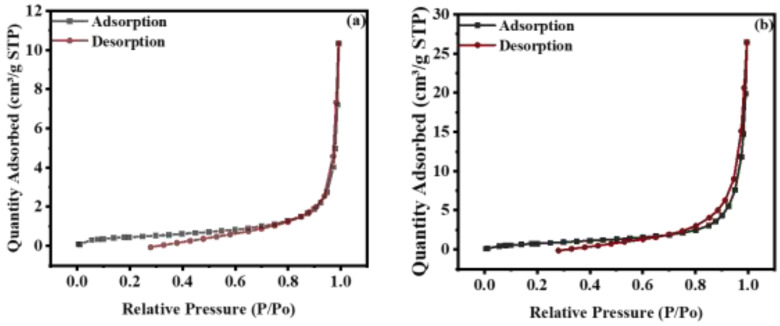
Nitrogen adsorption–desorption isotherm of (**a**) WO_3_ and (**b**) WO_3_/MXene nanocomposite.

**Figure 8 nanomaterials-12-00713-f008:**
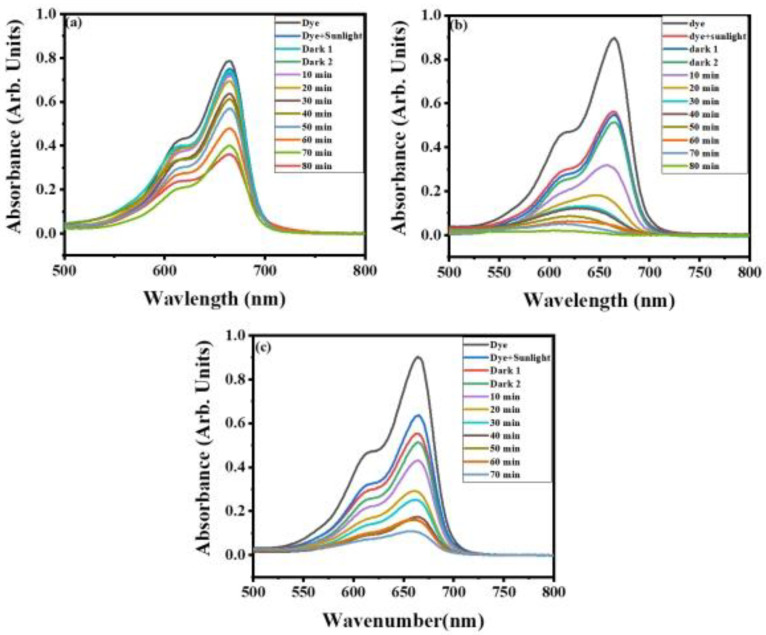
Absorption spectra for photocatalytic degradation of methylene blue by (**a**) MXene, (**b**) WO_3_, and (**c**) WO_3_/MXene nanocomposite.

**Figure 9 nanomaterials-12-00713-f009:**
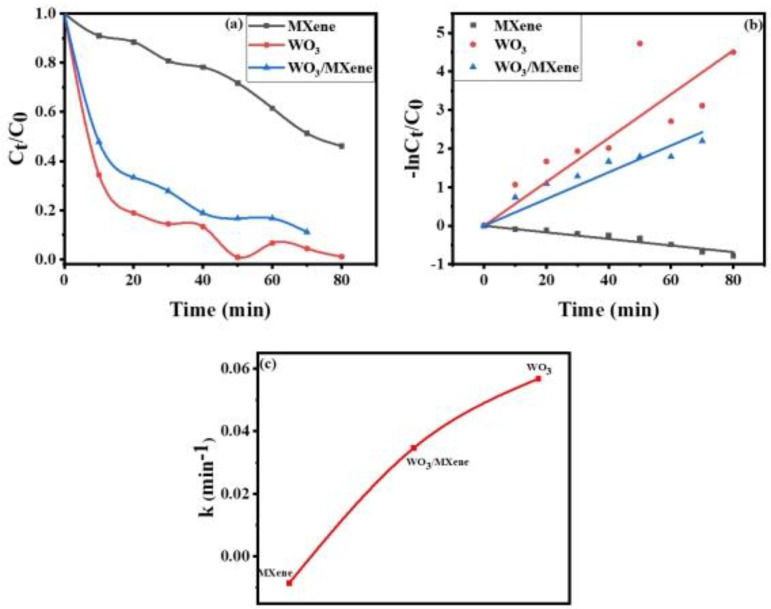
(**a**–**c**) Kinetic study for rate constant of MXene, WO_3_, and WO_3_/MXene nanocomposite.

**Figure 10 nanomaterials-12-00713-f010:**
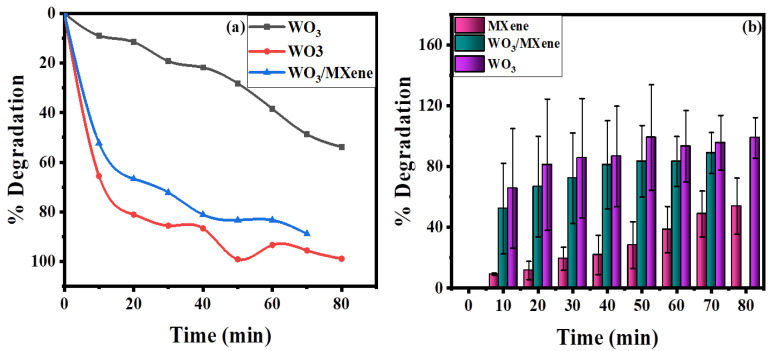
(**a**) Comparison of photocatalytic degradation of methylene blue by WO_3_, MXene, and WO_3_/Mxene (**b**) bar chart representation of percentage degradation with error bars.

**Figure 11 nanomaterials-12-00713-f011:**
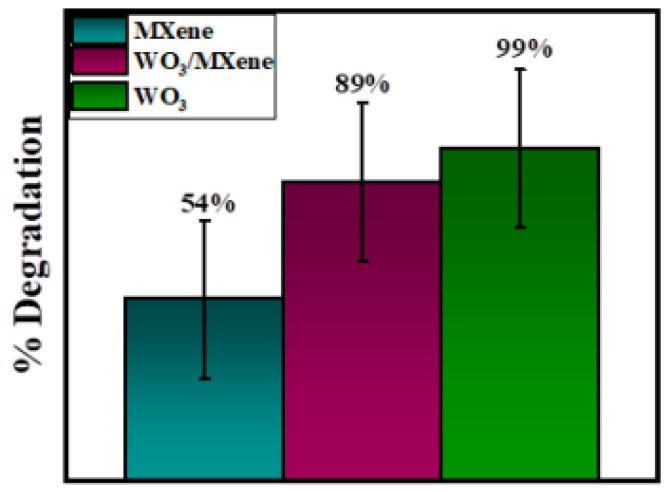
Comparison of degradation percentage of methylene blue using WO_3_, MXene, and WO_3_/MXene nanocomposite.

**Figure 12 nanomaterials-12-00713-f012:**
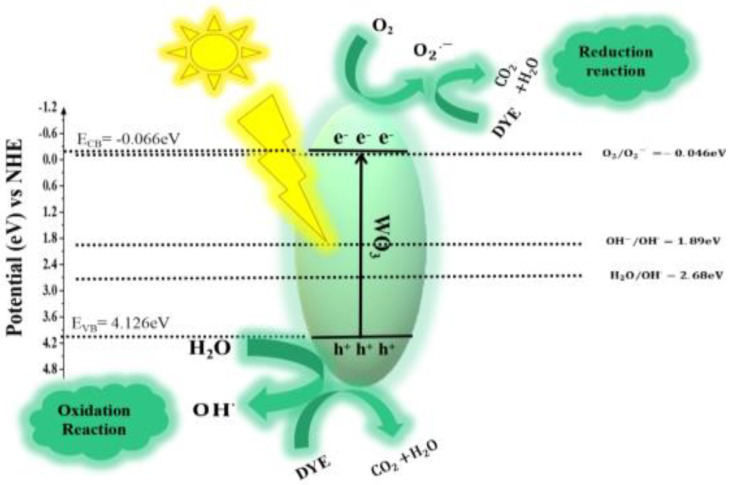
Z-scheme mechanism for the photocatalytic activity of WO_3_.

**Figure 13 nanomaterials-12-00713-f013:**
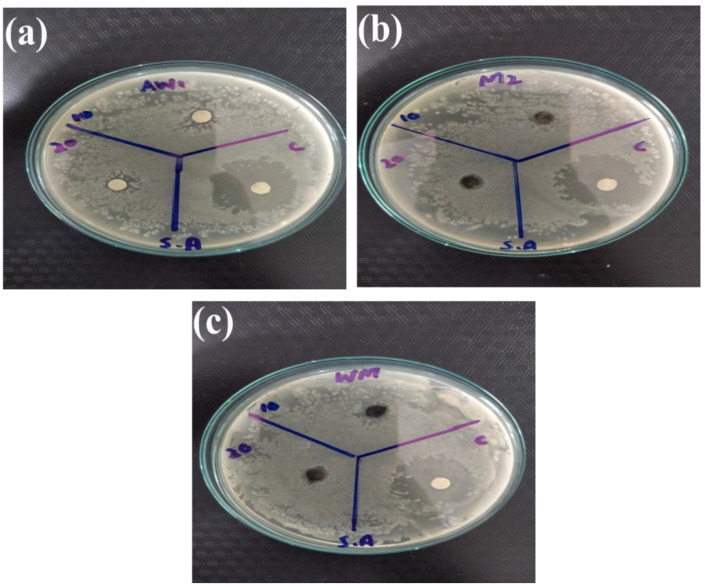
Antibacterial activity of (**a**) WO_3_, (**b**) Mxene and (**c**) WO_3_/Mxene against *Staphylococcus aureus*.

**Figure 14 nanomaterials-12-00713-f014:**
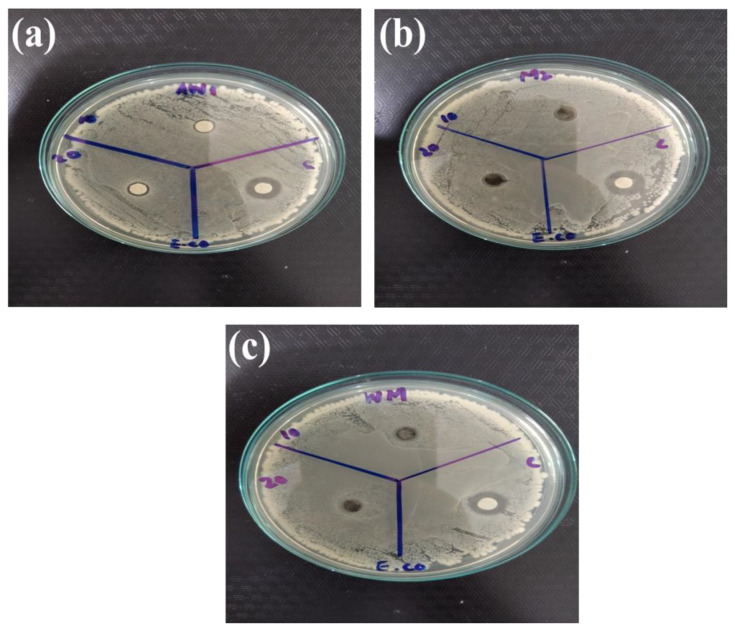
Antibacterial activity of as-prepared samples (**a**) WO_3_, (**b**) Mxene and (**c**) WO_3_/Mxene against *Escherichia coli (E. coli*).

**Figure 15 nanomaterials-12-00713-f015:**
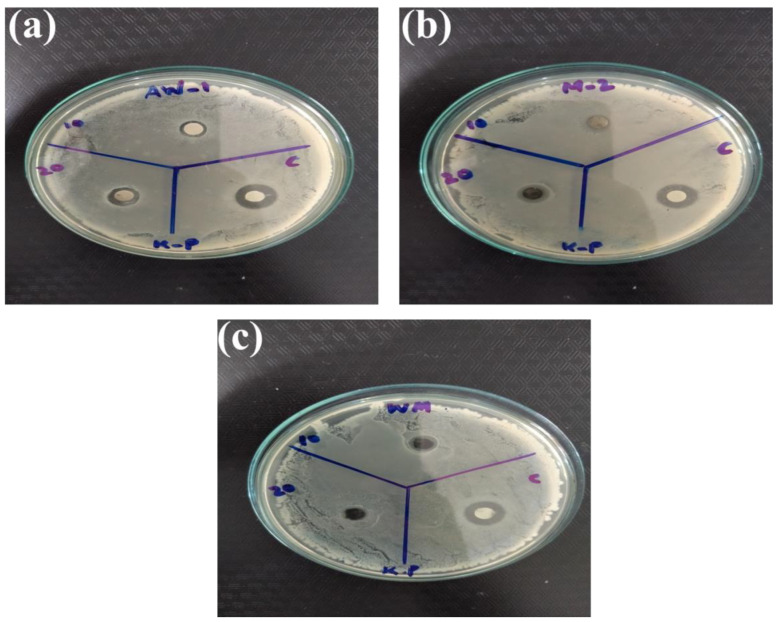
Antibacterial activity of as-prepared material (**a**) WO_3_, (**b**) Mxene and (**c**) WO_3_/Mxene against *Klebsiella pneumonia (K. pneumonia*).

**Figure 16 nanomaterials-12-00713-f016:**
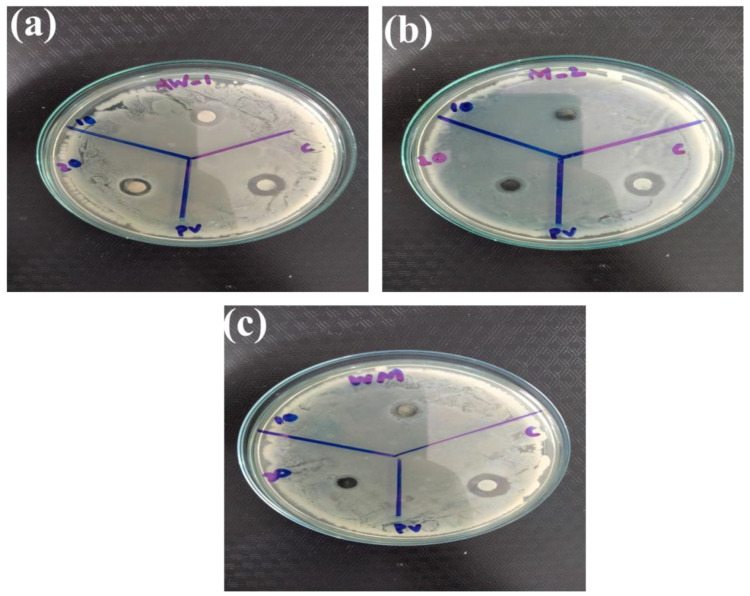
Antibacterial activity of as prepared materials (**a**) WO_3_, (**b**) Mxene and (**c**) WO_3_/Mxene against *Proteus vulgaris (P. vulgaris*).

**Table 1 nanomaterials-12-00713-t001:** Surface properties of WO_3_, WO_3_/MXene nanocomposite calculated from BET analysis.

S. No.	Properties	WO_3_	WO_3_/MXene Nanocomposite
1	BET surface area	1.63 m^2^/g	3.15 m^2^/g
2	Average particle size	3. 6 µm	1.9 µm
3	Total adsorption pore volume	0.0111 cm^3^/g	0.0307 cm^3^/g
4	Average pore width	27.3 nm	39 nm

**Table 2 nanomaterials-12-00713-t002:** Comparison between the degradation percentage of the current research with values reported in the literature.

Sr. No.	Photocatalyst	Pollutant	Light Source	% Degradation	Time (min)	Reference
1.	ZnO/Ag	Methyl orange	Visible light	78%	180	[62]
2.	NiO/Ag	Methyl orange	Visible light	42%	180	[62]
3.	TiO_2_/Ag	Methyl orange	Visible light	86%	180	[62]
4.	AgO	Methyl orange	Visible light	60.5%	50	[63]
5.	CoO	Methyl orange	Visible light	71.24%	50	[63]
6.	CdO	Methyl orange	Visible light	80.2%	50	[63]
7.	AgO–CoO–CdO/PACSGO	Methyl orange	Visible light	97.4%	50	[63]
8.	CdO	Methylene blue	Sunlight	78%	--	[64]
9.	CdO	Congo red	Sunlight	81%	--	[65]
10.	SnO_2_	Congo red	Sunlight	90%	--	[66]
11.	CdO	Alizarin red S	Sunlight	78%	--	[67]
12.	MgO	Alizarin red S	Sunlight	70%	--	[67]
13.	ZrO_2_	Methylene blue	UV-light	99%	--	[68]
14.	MgO	Methylene blue	Sunlight	88%	--	[69]
15.	NiO	Evans blue	Sunlight	88.13%	--	[70]
17.	CdO–NiO–ZnO	Methylene blue	Sunlight	89%	--	[71]
18.	CdO–ZnO–MgO	Methylene blue	Sunlight	91%	--	[72]
19.	CdO–MgO	Alizarin red S	Sunlight	82%	--	[73]
20.	WO_3_	Methylene blue	Sunlight	99%	80	Current Work
21.	WO_3_/MXene	Methylene blue	Sunlight	89%	70	Current Work
22.	MXene	Methylene blue	Sunlight	54%	80	Current Work

**Table 3 nanomaterials-12-00713-t003:** Results of antibacterial activity of as-synthesized samples.

Sr. No.	Sample	Concentration	*S. aureus*	*K. pneumoniae*	*E. coli*	*P. vulgaris*
1.	WO_3_/MXene	10 mg/mL	07 mm	08 mm	00 mm	00 mm
		20 mg/m	09 mm	00 mm	00 mm	00 mm
	Ciprofloxacin	10 ug/mL	22 mm	12 mm	12 mm	13 mm
2.	MXene	10 mg/mL	00 mm	00 mm	08 mm	00 mm
		20 mg/m	08 mm	10 mm	08 mm	10 mm
	Ciprofloxacin	10 ug/mL	22 mm	12 mm	12 mm	13 mm
3.	WO_3_	10 mg/mL	07 mm	07 mm	06 mm	00 mm
		20 mg/m	07 mm	10 mm	07 mm	11 mm
	Ciprofloxacin	10 ug/mL	22 mm	12 mm	12 mm	13 mm

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
