# Peer review of "Synthesis, Characterization, Photocatalysis, and Antibacterial Study of WO3, MXene and WO3/MXene Nanocomposite"

_nanomaterials, 2022, doi:10.3390/nano12040713_

Round 1

Reviewer 1 Report

Warsi et al reported Synthesis, Characterization, photocatalytic and anti-bacterial study of WO3, MXene and WO3/MXene nanocomposite. The synthesized tungsten oxide, MXene and its composite were characterized by X-ray diffraction (XRD), scanning electron microscopy (SEM), Fourier transform infrared (FTIR), Energy Dispersive X-Ray Analysis (EDX), and Brunauer–Emmett–Teller (BET). The authors should improve their work with the following comments:

  1. why this nanocomposite have antibacterial activity, what is its mechanism?
  2. please discuss the following papers in your introduction to give a better outlook of your work (10.1016/j.ajps.2020.11.004, 10.3390/nano11102535, https://doi.org/10.1016/j.ijbiomac.2021.11.028)
  3. some figures have low quality, please improve the resolution (1, 2, 4, 11, 12)
  4. subscripts and superscripts in incorrect in the whole manuscript for example page 2 lines 77, 80, etc…
  5. the manuscript need a complete English editing
  6. how the samples prepared for SEM analysis. Please completely add
  7. what was the crystalline size of the samples in XRD?
  8. Authors should discuss cytotoxic effect of their nanocomposite on environment

Reviewer 2 Report

The semiconductor nanoparticles have been studied for different photocatalysis efficiency such as organic pollution degradation, and toxic metal removal. Since MXene, as a narrow bandgap semiconductor photo catalytically degrades dyes under visible light, the photocatalytic application of MXene has been deeply studied. On the other hand, tungsten trioxide is the potent semiconductor stable for the fabrication of photocatalysis systems at room temperature. However, WO3 has poor charge transport for photo-generated e/h+ pairs, due to the wide-bandgap. Then, the best method for modification of photocatalysis efficiency of the semiconductor oxide is combining with the nano semiconductor like Mxene. Hybrid photocatalyst (Mxene/WO3) has been reported. The concept of having hybrid material itself is not new but the chosen advanced material like Mxene could be new.

The manuscript is OK-ish with supportive physical and practical validations; it can be considered after revision.

Following are my specific comments:

Introduction

  • The abstract should state the performance characteristics (providing values) along with their efficiency.
  • “Meanwhile, Fe2O3 and ZnO are quite unstable in wastewater” need a bit more explanation.
  • Next sentence; “metal sulfides…..” this is not true. Cu2MoS4 is extensively applicable in photochemical and electronic devices.
  • “treating cancer by tungsten oxide in mice” either delete or require more information to support this statement.
  • Tungsten oxide has also been shown to be synthesized by other facile approaches include (doi.org/10.3390/nano11030580).
  • such as SCs, - explain (supercapacitors)
  • Please provide the structural formula for Mxene.

Experimental

  • Section 2.6 needs to be appropriately referenced back to the literature relating to the degradation, adsorption/desorption, etc.
  • Compared with other adsorbent materials, why Mxenes are so attractive for heavy metals in the industry?
  • Was the antibacterial test duplicated (replicated) for reproducibility?

Results and Discussion

  • Section 3,1 please provide the standard JCPDS card for peak references.
  • Section 3.1 Below Fig. 3, second line AND should read as and.
  • What sort of information has been inferred from the SEM images? The particle size needs to be quoted.
  • The nanosheet-like hierarchical porous structure associated with a large surface area is similar to the one reported for other metal oxides (10.1039/C8NR03824D). Please cite the reference and discuss.
  • In Figure 5, please provide the band wavenumbers.
  • Section 3.4: The photocatalytic response depicts that the presence of WO3 nanostructure and Mxene improved the charge transportation and reduces the recombination rate of e−/h+; which needs to be explained.
  • The obtained degradation can be benchmarked with other metal oxide composites reported.
  • All the equations need to be numbered and the variables used need to be defined.
  • Section 3.5: Please provide the level of toxicity that has been decreased while using Mxene composites.

Reviewer 3 Report

The manuscript “Synthesis, Characterization, photocatalytic and anti-bacterial study of WO3, MXene and WO3/MXene nanocomposite” deals with the preparation of composites with biological and photocatalytic properties. The work is well organized and written. The publication is recommended after some revisions, as follows:

- Abstract. Add antibacterial quantitative results to this section.

- Introduction. The state of the art related to the production of antibacterial materials can be enlarged, adding recent works that describe different preparation methods. As an example, see this work: Baldino et al., Production, characterization and testing of antibacterial PVA membranes loaded with HA-Ag3PO4 nanoparticles, produced by SC-CO2 phase inversion, Journal of Chemical Technology and Biotechnology, 2019, 94(1), pp. 98–108; etc…

- Results. The quality of SEM images and of the various figures is too low. Improve. Error bars are not present in the diagram of Figure 9.

Author Response

Please see the atatchment.

Round 2

Reviewer 1 Report

It can be accepted in the current format

Author Response

Already submitted

Reviewer 2 Report

I went through the revised part of the manuscript along with the responses to the reviewer's queries. The authors have substantially improved the quality and discussion of the work. In this reviewer's opinion, it is suitable to publish now.

Author Response

Already submitted